# Prevention of Surgery-Induced Dry Eye by Diquafosol Eyedrops after Femtosecond Laser-Assisted Cataract Surgery

**DOI:** 10.3390/jcm11195757

**Published:** 2022-09-28

**Authors:** Kenichiro Yamazaki, Junko Yoneyama, Ryuta Kimoto, Yuko Shibata, Tatsuya Mimura

**Affiliations:** 1Department of Ophthalmology, Omiya Nanasato Eye Institute, Saitama 330-0017, Japan; 2Department of Ophthalmology, Teikyo University, Tokyo 173-8605, Japan

**Keywords:** diquafosol, dry eye, femtosecond laser-assisted cataract surgery

## Abstract

Purpose: To analyze the protective effects of diquafosol eyedrops on the ocular surface following femtosecond laser-assisted cataract surgery (FLACS). Design: A prospective, randomized contralateral study. Methods: Bilateral FLACS with a trifocal IOL (PanOptix) implantation was performed in 40 eyes in 20 patients (10 males, 10 females, average age 68.8 ± 6.3 years old). Patients received 3% diquafosol eyedrops six times daily in one randomly chosen eye (diquafosol group), and physiological saline six times a day in the other eye (control group). Other medication included 1.5% levofloxacin, 0.1% dexamethasone and 0.1% diclofenac three times daily in both eyes. The pre and post-operative tear break-up time (BUT), superficial punctate keratopathy (SPK) scores and visual function were compared between both eyes, and all patients answered the dry-eye-related quality of life score (DEQS) questionnaire. Results: The BUT between groups was similar pre-operatively and on the first day post-op; however, the BUT was statistically longer in the diquafosol group compared to saline at 1 week (5.5/3.7 s) and 2 weeks (4.8/3.0 s) (*p* < 0.05). There was no difference in the SPK score, best corrected distance visual acuity, tear meniscus height, contrast sensitivity, DEQS and Schirmer test at all time points. Spherical aberration was statistically lower in the diquafosol group at 1 week. The protective effects of diquafosol on the BUT was more pronounced in patients with a pre-operative BUT of less than 5 s compared with those with a BUT longer than 6 s. Conclusions: Diquafosol eyedrops prevented the shortening of the BUT following FLACS, even in patients with short pre-operative BUT values.

## 1. Introduction

Dry eye is caused by multiple factors and results in symptoms of visual disturbance and discomfort [1]. Diquafosol ophthalmic solution 3% (Diquas^®^) is a novel dry eye drug currently approved in Japan and South Korea. Diquafosol is reported to promote tear fluid and mucin secretion through purinergic P2Y_2_ receptor agonist action [2,3]. Diquafosol is a synthetic derivative of the naturally occurring nucleotide uridine 5′-triphosphate, a ligand of the P2Y_2_ receptor, which is expressed in goblet cells and meibomian glands in the ocular surface [4]. P2Y_2_ receptor agonists stimulate mucin secretion from goblet cells in vitro. Mucin stimulants are a new target for the treatment of dry eye since mucin increases the tear film stability and protects against the desiccation of the ocular surface [5]. Diquafosol facilitates fluid transport from the serosal to the mucosal side via chloride channel activation following the elevation of the calcium ion concentration in epithelial cells on the ocular surface through the P2Y_2_ receptor [5,6]. Diquafosol stimulated fluid and Cl^−^ transport across the rabbit conjunctival epithelium in the serosal to mucosal direction. Both tear fluid and active Cl^−^ secretion from isolated rabbit conjunctiva were stimulated by diquafosol in a concentration-dependent manner [6,7].

A randomized, double-blind, multicenter clinical trial revealed that the diquafosol solution significantly decreases fluorescein corneal and rose bengal scores compared with artificial tears [8]. Diquafosol significantly improved the tear film breakup compared to sodium hyaluronate 0.1% by increasing the tear volume and stimulating mucin secretion by secretary goblet cells [5,7,9,10,11]. Diquafosol has also been reported to upregulate the expression of membrane-bound mucin genes in corneal epithelial cells [12]. This action leads to an increase in tear fluids on the ocular surface for up to 30 min in normal human eyes after a single dose [11], in addition to promoting tear film in the ocular surface epithelium [7,9,12,13].

A diquafosol ophthalmic solution of 3% demonstrated efficacy in dry eye due to various causes, including dry eye following cataract surgery. [2,14,15] The dry eye prevalence after surgery decreased compared to before cataract surgery by using diquafosol postoperatively. The break-up time (BUT) was shortened significantly, and the fluorescein staining score also increased significantly at 4 weeks after surgery, but diquafosol application significantly improved the ocular surface subjective symptom scores. (*p* < 0.001) [16]. The BUT, Ocular Surface Disease Index (OSDI) and Oxford scheme score showed significant improvement at one and three months post-surgery [17].

Cataract surgery has been reported to increase the incidence of dry eye post-operatively by up to 8.4% [18,19]. In the cases of pre-existing dry eye, the tear production and tear breakup time (TBUT) decreased after cataract surgery, leading to ocular discomfort and irritation [20]. Compared with the group of patients with no history of dry eye, patients with pre-existing dry eye showed significantly higher ocular symptom scores, a shorter BUT and higher lid margin abnormalities, meibum quality and expressibility after cataract surgery [21,22].

Surgery-induced dry eye leads to patient dissatisfaction following multifocal intraocular lens (IOL) surgery, and one study reported that the use of cyclosporine 0.05% therapy reduced dry eye signs and improved visual quality after surgery. However, cyclosporin is not approved in Japan for dry eye. In fact, up to 35% of complaints following multifocal IOL surgery was attributed to dry eye, and the most common interventions pursued were treatment for dry eye (24%) [23]. There are no studies to date that show the efficacy of using diquafosol following multifocal IOL surgery to improve dry eye and visual function.

Compared to manual cataract surgery, femtosecond laser-assisted cataract surgery results in a higher frequency and severity of dry eye [24]. Both surgical methods worsened dry eye postoperatively; however, FLACS may have a higher risk of surgery-induced dry eye. Patients with pre-existing dry eye who had FLACS had more severe dry eye symptoms and a worse visual function than those after manual surgery [25]. The aim of this study was to prospectively analyze the protective effects of diquafosol eyedrops on the ocular surface following FLACS.

## 2. Patients and Methods

This was a prospective, randomized, contralateral study involving patients with cataracts who underwent bilateral phacoemulsification with FLACS and diffractive trifocal IOLs implantation in February 2020. The study was approved by the Ethics Committee at Omiya Nanasato Eye Institute (approval number: D017-N-O32, approval date: 1 February 2020). The study was registered in a clinical trial registry (UMIN000039491; date of registration: 15 February 2020). Patients who were 50 years or older and had cataracts in both eyes were eligible for the study. All patients signed a written consent form prior to enrollment. Patients were excluded from the study if they had previously used diquafosol or topical steroids. Patients who did not have an indication for multifocal IOLs, and patients with co-morbidities such as glaucoma age-related macular degeneration (AMD) or diabetic retinopathy, were also excluded from the study. Visual acuity, SPK scores and BUT were similar between the diquafosol and control eyes (Appendix A).

All patients were scheduled to undergo FLACS cataract surgery in one eye, and then the other eye 1 week later. Patients were randomized to receive 6 daily doses of topical 3% diquafosol in 1 eye, and physiological saline solution in the other eye starting on the first day after cataract surgery for a period of 2 weeks. Both the choice of eye to be treated with diquafosol and the eye to first undergo surgery was randomly chosen. Eye drop bottles were masked so that the patient was not aware of which medication was used. Other post-operative treatment included 1.5% levofloxacin, 0.1% dexamethasone and 0.1% diclofenac eye drops in both eyes, 3 times daily for up to 2 weeks. Levofloxacin and 0.1% diclofenac were also applied 3 times on the day of surgery, and 0.5% tropicamide was used for mydriasis.

FLACS and phacoemulsification and aspiration (PEA) were carried out by the same surgeon (KY) in the same institution. FLACS was carried out using the LenSx system to perform 2 corneal incisions, capsulotomy and lens fragmentation. Corneal wound sizes were 2.3 mm for the primary incision and 1.2 mm for the secondary incision. FLACS and PEA were performed in the same operating room. Additional topical anesthesia (4% lidocaine) and disinfection with 0.5% povidone iodine were conducted between FLACS and PEA. PEA was performed using the Centurion^®^, with implantation of the diffractive trifocal IOL TFNT00 for eyes with less than −1.0 D astigmatism, and TFNT30, 40, 50 or 60 depending upon the degree of astigmatism greater than −1.0 D. Both eyes were examined at 1 month pre-op and 1 day, 1 week and 2 weeks post-op. Uncorrected distance visual acuity (UDVA) and corrected distance visual acuity (CDVA) at 5 m, uncorrected near visual acuity (UNVA) and corrected near visual acuity (CNVA) at 40 cm, uncorrected intermediate visual acuity (UIVA) and corrected intermediate visual acuity (CIVA) at 70 cm, BUT, SPK scores, Schirmer test values and spherical aberration using the CASIA 2 (Tomey) were recorded at each time point. Visual acuity was expressed in logMAR. Contrast sensitivity was measured pre-op and 2 weeks post-op. Using the CASIA 2, tear meniscus height, area, length and circumference was measured at 1 day, 1 week and 2 weeks after surgery.

BUT was measured by slit lamp with a blue filter using a standard fluorescein paper strip and balanced saline solution. The Schirmer test was conducted using topical anesthesia. SPK was scored using the National Eye Institute (NEI) staining grade. In brief, the cornea was divided into five sections, each assigned a value from 0 (no staining) to 3 (severe staining) based on the amount, size and confluence of the punctate staining. Contrast sensitivity was measured with the CSV-1000 (Vector Vision), and the dry-eye-related quality of life score (DEQS) questionnaire was performed at the end of the 2 week study period. Both the BUT and SPK scores were recorded by 2 independent examiners (RK and YS) who were not told to which group the eye belonged.

## 3. Statistical Analysis

A statistical power analysis was performed to estimate the number of cases required to observe a difference of 20%, confidence level at 95% and a standard deviation of 20. The calculations revealed that the minimum number of cases required to show a significant difference between the two groups was 16. The two-tailed paired or unpaired Student’s t-test was used to compare the mean values between two groups. Relations among variables were investigated by calculating the two-tailed Pearson correlation coefficients and partial correlation coefficients. Data were expressed as mean (standard deviation) or percentages. Statistical analyses were performed with SAS System software version 9.1 (SAS Institute Inc., Cary, NC, USA), and significance was accepted at *p* < 0.05. Normality and chi-square tests were also performed in this study. For the normality test, the mean (standard deviation) was used.

## 4. Results

This study included 40 eyes of 20 patients (10 male, 10 female). The average age of the patients was 68.8 ± 6.3 years old (range 60 to 82 years old). None of the patients had moderate to severe dry eye prior to surgery. Patients with a short BUT had mild superficial punctate keratitis (SPK). The pre-operative best corrected visual acuity was 0.42 ± 0.12 logMAR in the diquafosol (DQS) group and 0.41 ± 0.11 in the control group.

The average laser irradiation time was 33.4 ± 4.0 s, and the average phacoemulsification time was 10.6 ± 3.5 min. The cumulative dispersed energy (CDE) of the Centurion device was 3.3 ± 1.9. The Emery–Little classification grade of cataracts ranged between G2 and G4, and there was no difference in cataract grade between the DQS group and control. There were no incidences of posterior capsule rupture, IOL decentration or endophthalmitis during the study. Regarding the comparison of the post-operative best corrected visual acuity between the DQS group and control at 5 m, 70 cm and 40 cm, there was no statistical difference at all time points measured post-operatively (Figure 1A–C).

The spherical aberration measured using the CASIA 2 was 0.24 ± 0.18/0.24 ± 0.16 one day post-op, 0.23 ± 0.12/0.27 ± 0.11 one week post-op and 0.28 ± 0.11/0.32 ± 0.13 2 weeks post-op in the DQS group and control, respectively. Spherical aberration was statistically less in the DQS group at 1 week post-op, but not in any of the other time points (Figure 1D). The BUT in the DQS/control group was 4.9 ± 2.3/4.8 ± 2.5 s preoperatively, 4.1 ± 3.4/4.2 ± 3.2 s one day post-op, 5.5 ± 3.6/3.7 ± 3.3 s 1 week post-op and 4.8 ± 2.8/3.0 ± 1.8 s at 2 weeks post-op, respectively. Although there was no statistical difference pre-op and one day post-op, the BUT was statistically longer (*p* < 0.05) in the DQS group at 1 week and 2 weeks post-op (Figure 2A).

The SPK score in the DQS/control group was 0.51 ± 0.76/0.42 ± 0.81 preoperatively, 1.25 ± 1.16/1.21 ± 1.15 one day post-op, 1.25 ± 1.36/0.83 ± 1.02 1 week post-op and 0.70 ± 1.06/0.62 ± 0.99 2 weeks post-op, respectively. There was no statistical difference between both groups.

There was no difference in Schirmesr I test scores between DQS (5.3 ± 3.6 mm) and the control (4.6 ± 4.1 mm) (Figure 2B).

The tear meniscus area measured by CASIA 2 was 0.046 ± 0.035/0.046 ± 0.031 mm^2^ one day post-op, 0.035 ± 0.038/0.032 ± 0.013 mm^2^ 1 week post op and 0.066 ± 0.011/0.005 ± 0.065 mm^2^ 2 weeks post-op in the DQS and control group, respectively. There was also no difference in tear meniscus height and circumference (Figure 2C).

Thirty eyes had a pre-operative BUT shorter than 5 s, whereas 10 eyes had a pre-operative BUT longer than 6 s. Sixteen eyes of pre-operative short BUT eyes were in the DQS group, and 14 eyes were in the control group. When both pre-operative short BUT eyes and long BUT eyes were analyzed separately, the short BUT group showed a statistically significant difference in BUT after 2 weeks, where the DQS group improved the BUT compared to the control (Figure 2D). There was no difference in eyes with a pre-operative long BUT. There was no difference in contrast sensitivity between the DQS group and control throughout the study.

The DEQS questionnaire did not reveal a statistical difference in any of the questions; however, there was a marginal difference in the question “Felt down due to eye symptoms”, which was higher in the control group (0.6 ± 1.3) compared to the DQS group (0.3 ± 0.7) (*p* = 0.07). A higher score implies more discomfort. There were no side effects observed in any of the eyes enrolled in the study.

## 5. Discussion

Post-surgical dry eye is one of the leading causes of patient dissatisfaction following multifocal IOL surgery [23]. Phacoemulsification reduces tear film stability due to several mechanisms, including the frequent application of eye drops, mechanical damage, peripheral nerve injury at the incision, exposure of the ocular surface to hard light and repeated washing with lavage fluid [3,7]. Topical diclofenac applied following surgery was reported to be responsible for the decrease in conjunctival goblet cell density [26]. One report showed that cyclosporin eye drops were effective in treating dry eye following multifocal IOL surgery [27]. Unfortunately, cyclosporin drops are not available universally, so we examined the protective effects of diquafosol in preventing post-surgical dry eye after multifocal IOL surgery. 

One study reported higher corneal fluorescein staining and ocular surface disease index scores following FLACS compared to manual phacoemulsification due to the use of a suction ring [24]. While these changes were only during the early post-operative period, the authors concluded that the difference was due to the suction ring used in FLACS. The use of a suction ring during LASIK has also been reported to induce dry eye by decreasing the goblet cell density, thereby disrupting the mucin layer [28]. LASIK-induced dry eye was reported to be less severe after femtosecond LASIK than after microkeratome LASIK, further suggesting the role of a suction ring in post-surgical dry eye [29]. The LenSx system used in this study uses a soft contact lens (SCL) for the suction ring, and, since the entire procedure with the suction ring is approximately 1 min, the effects of the suction in FLACS may be less severe than LASIK.

Dry eye signs and symptoms following FLACS seems to be most severe during the first week, and most signs recover to pre-operative levels by 3 months [30]. While we did not observe a difference in fluorescein staining, we found that the use of diquafosol following FLACS resulted in a longer BUT in this study, suggesting that mucin production by goblet cells was maintained by diquafosol, thus preventing excessive tear evaporation. This may also explain improved symptoms in the DQS group. Reports have shown that patients with pre-existing dry eye suffer from an exacerbation of symptoms following FLACS surgery [24,25]. We found that diquafosol was effective in improving the BUT at 1 w and 2 w when all cases were analyzed, but when limited to cases with pre-operative BUT values of less than 5 s, the protective effect was not observed until 2 w. This suggests that patients with pre-operative dry eye may benefit from the pre-operative application of diquafosol, or a longer treatment period after surgery.

An interesting observation in our study was that spherical aberration improved using diquafosol, especially at 1 week after surgery. This suggests that an increased aberration may be a major factor in patient dissatisfaction following multifocal IOL surgery. Since expectations and costs are higher with multifocal IOLs, patient dissatisfaction may be more pronounced compared to mono-focal IOLs. Another study reported that dry eye patients after cataract surgery had a visual dysfunction due to higher order aberrations (HOAs) [31]. They also found that diquafosol was effective in treating dry eye in order to improvement visual function. While other topical drugs are available, diquafosol was shown to significantly lower OSDI scores, extend the BUT time and lower epithelial lesions compared to sodium hyaluronate eye drops [32]. Interestingly, they also found an improvement of spherical aberrations with diquafosol at 4 weeks after surgery, which is similar to our results. Changes in HOAs suggest that dissatisfaction may not be measured by visual acuity alone, since we did not find a difference in post-operative far, intermediate (70 cm) or near (40 cm) uncorrected visual acuity in our study.

The short follow-up period of 2 weeks may be a limitation of this study. However, the objective of this study was to observe the effects of diquafosol on short-term dry eye symptoms due to surgery, which may affect patient satisfaction. A recent article showed that pre-operative treatment for dry eye significantly improved the corneal surface condition and accuracy of the predicted postoperative refraction in dry eyes [33]. The number of subjects enrolled was also relatively small. A larger study with a longer follow up period is required to further confirm the long-term effectiveness of diquafosol drops to prevent post-operative dry eye following FLACS. 

## 6. Conclusions

In conclusion, we found that the use of diquafosol is effective in preventing post-operative dry eye following FLACS. Diquafosol seems to work by compensating for mucin production by goblet cells that may have been damaged by the suction ring used during surgery. Of all of the dry eye parameters studied, the BUT time was significantly improved by diquafosol, which may be related to the decrease in HOAs.

## Figures and Tables

**Figure 1 jcm-11-05757-f001:**
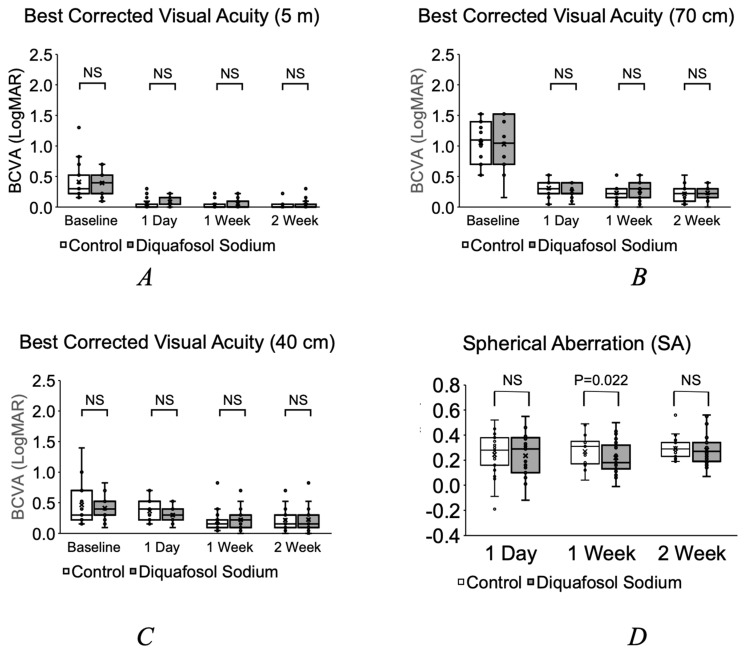
Pre-operative and post-operative best corrected visual acuity at 5 m (**A**), 70 cm (**B**) and 40 cm (**C**) and post-operative spherical aberration measured by CASIA 2 (**D**). NS = not significant.

**Figure 2 jcm-11-05757-f002:**
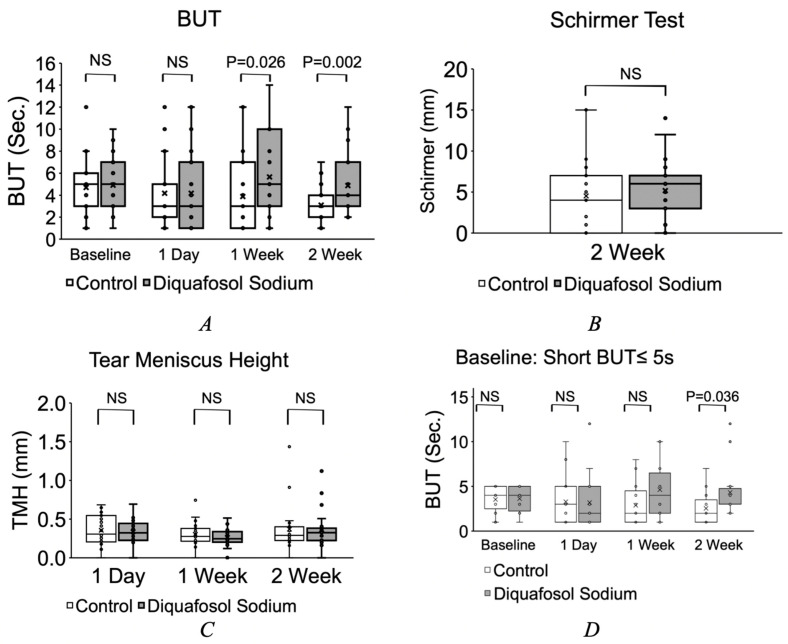
Post-operative evaluations of the ocular surface (**A**–**C**) in all eyes, and in eyes that had short BUT pre-operatively (**D**). NS = not significant.

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
