# Peer review of "Prevention of Surgery-Induced Dry Eye by Diquafosol Eyedrops after Femtosecond Laser-Assisted Cataract Surgery"

_jcm, 2022, doi:10.3390/jcm11195757_

Round 1

Reviewer 1 Report (Previous Reviewer 3)

The authors have addressed all changes and suggestions correctly

Author Response

Comment: The authors have addressed all changes and suggestions correctly

Response: Thank you for reviewing our manuscript. We appreciate your comment.

Reviewer 2 Report (New Reviewer)

Prevention of Surgery-Induced Dry Eye by Diquafosol Eyedrops after Femtosecond Laser-Assisted Cataract Surgery

Manuscript review

Summary

This small scale randomized contralateral study brings to light the benefits of using Diquafosol following FLACS surgery, in terms of tear break up time and regarding the spherical aberration. This study benefits from a strong and rigorous design, which supports the findings and the benefits of this molecule against dry eye following FLACS cataract interventions.

General concept comments

The article is well organized, comprehensive, on a relevant topic. No author of the present manuscript appears as authors of any cited work in the manuscript. 

In the recent years, several studies regarding Diquafosol and cataract surgery have been published (Jun I et al. Effects of Preservative-free 3% Diquafosol in Patients with Pre-existing Dry Eye Disease after Cataract Surgery: A Randomized Clinical Trial; Baek J et al. The Effect of Topical Diquafosol Tetrasodium 3% on Dry Eye After Cataract Surgery; Kim S et al. A randomised, prospective study of the effects of 3% diquafosol on ocular surface following cataract surgery; Park DH et al. Clinical Effects and Safety of 3% Diquafosol Ophthalmic Solution for Patients with Dry Eye After Cataract Surgery: A Randomized Controlled Trial; Cui L et al. Effect of diquafosol tetrasodium 3% on the conjunctival surface and clinical findings after cataract surgery in patients with dry eye.). 

However, the focus on FLACS and its effects on the ocular surface and Dry Eye Disease represents an important element of novelty. 

The results are clearly described and discussed; however, I would recommend looking into expanding the Tables and Figures, in order to better support the understanding of the data.

The references are reasonably recent - 9/33 citations are from the last 5 years. The English usage is appropriate and understandable, and the Ethics statements are adequate.

Specific comments (Suggested modifications)

  • Patients and methods - it should be stated that the visual acuity is expressed in LogMAR

  • I suggest defining abbreviations at the first appearance in the text (the CSV device, IOL, AMD)

  • Line 92 - I suggest a clearer statement of the inclusion criteria in the study

  • Line 130 - I suggest more details on the National Eye Institute Staining Grade (the areas, the scoring method, etc.)

  • Line 148 - I suggest stating the visual acuity before the intervention - in the text as well, not only in the figures and supplementary table

  • Line 189 - there were 40 eyes in total - here 13 and 33 (31 in the mentioned table) eyes are reported - has the BUT been calculated in certain patients several times? What is the explanation?

Recommendation

I believe this is a strong article that contributes greatly to the general knowledge in this domain. More importantly, I believe this paper is of wide interest to the readers of the Journal and fits the scope. I recommend accepting the paper, after minor revisions, detailed previously.

I congratulate the authors for their efforts and look forward to their revisions.

Author Response

To Reviewer 2

Thank you for your constructive comments to improve the quality of our paper.

General comment: I would recommend looking into expanding the Tables and Figures, in order to better support the understanding of the data.

Response: We enlarged all figures for clarity. Please let me know if this was not the intent of your comment.

Comment: Patients and methods - it should be stated that the visual acuity is expressed in LogMAR

Response. The following sentence was added in the text as recommended: "Visual acuity is expressed in logMAR ".  (Line 127)

Comment: I suggest defining abbreviations at the first appearance in the text (the CSV device, IOL, AMD)

Response: Thank you for your comment. We defined abbreviations for intraocular lens (IOL) in line 75, and age-related macular degeneration (AMD) in line 99.

CSV-1000 is a product name, so I added manufacturer's name (Vector Vision) in line 135.

I also defined diquafosol (DQS) in line 157.

Comment: Line 92 - I suggest a clearer statement of the inclusion criteria in the study

Response: Thank you for your suggestion. The following sentence on inclusion criteria was added in the text: "Patients who were 50 years or older and had cataracts in both eyes were eligible for the study." (Line 95 -96).

Comment: Line 130 - I suggest more details on the National Eye Institute Staining Grade (the areas, the scoring method, etc.)

Response: Additional details of National Eye Institute Staining Grade was added in the text (line 132 – 134).

Comment: Line 148 - I suggest stating the visual acuity before the intervention - in the text as well, not only in the figures and supplementary table

Response: We added pre-operative best corrected visual acuity. (Line 156 - 157)

Comment: Line 189 - there were 40 eyes in total - here 13 and 33 (31 in the mentioned table) eyes are reported - has the BUT been calculated in certain patients several times? What is the explanation?

Response: We apologize for the confusion. This is an error that occurred when re-writing the manuscript from an older version which included patients with pre-diagnosed dry eye, which is an exclusion criteria and should not have been included in the analysis.

The error in case number was corrected the numbers in the main manuscript (line 193 -198) and figure 2, D. I also explained that 16 eyes in the DQS group (16 eyes) and 14 eyes in the control group had pre-operative short BUT for clarity.

This manuscript is a resubmission of an earlier submission. The following is a list of the peer review reports and author responses from that submission.

Round 1

Reviewer 1 Report

Introduction 

Line 44: Please correct spelling of "phopshate" to "phosphate".

Line 55: Please change "trials" to "trial".

Line 67 & 69: Please define BUT and OSDI upon first use in the body of the manuscript.

Line 74: Please delete "group".

Lines 86 - 89: I suggest rewriting this as "Compared to manual cataract surgery, femtosecond laser-assisted cataract surgery results in higher frequency and severity of dry eye."

Methods

Lines 98-99: Trifocal multifocal IOL seems redundant. "Trifocal IOL" should be sufficient as trifocal implies multifocality. 

Lines 105-106: This statement is a bit confusing. It creates the impression there were two separate groups: one receiving diquafosal and the other receiving control substance. Please modify statement to indicate the comparison is between treated and control eye. With the study design used, it is a given that pre-enrollment age will be same between treated and control eyes. 

ne 118: Please define PEA before using the abbreviation. This is to help readers who are naive to cataract surgery understand what it means.

Line 128: Please define all abbreviations that are being used for the first time.

Line 129: Please change to "was" to "were".

Lines 134 - 136: Please reconstruct for clarity.

Author Response

Introduction

Line 44: Please correct spelling of "phopshate" to "phosphate".

Response: We corrected the misspelling as suggested.

Line 55: Please change "trials" to "trial".

Response: The grammar was corrected as suggested.

Line 67 & 69: Please define BUT and OSDI upon first use in the body of the manuscript.

Response: BUT and OSDI was defined upon first use in the manuscript.

Line 74: Please delete "group".

Response: "Group" was deleted from the text.

Lines 86 - 89: I suggest rewriting this as "Compared to manual cataract surgery, femtosecond laser-assisted cataract surgery results in higher frequency and severity of dry eye."

Response: The text was rewritten as suggested.

Methods

Lines 98-99: Trifocal multifocal IOL seems redundant. "Trifocal IOL" should be sufficient as trifocal implies multifocality.

Response: "Trifocal multifocal IOL" was changed to "Trifocal IOL" as suggested.

Lines 105-106: This statement is a bit confusing. It creates the impression there were two separate groups: one receiving diquafosal and the other receiving control substance. Please modify statement to indicate the comparison is between treated and control eye. With the study design used, it is a given that pre-enrollment age will be same between treated and control eyes.

Response: Thank you for the valuable comment. In order to be more clear that the study was a comparison between eyes of the same patient, we changed the sentence as follows: "Visual acuity, SPK scores and BUT were similar between the diquafosol and control eyes".

Line 118: Please define PEA before using the abbreviation. This is to help readers who are naive to cataract surgery understand what it means.

Response: We defined PEA as phacoemulsification and aspiration in the text.

Line 128: Please define all abbreviations that are being used for the first time.

Response: We defined UDVA, CDVA, UNVA, CNVA, UIVA, and CIVA upon first use in the manuscript.

Line 129: Please change to "was" to "were".

Response: "Was" was changed to "were" as suggested. We apologize for the grammatical error.

Lines 134 - 136: Please reconstruct for clarity.

Response: The section was re-written as follows; "SPK was scored using the National Eye Institute Staining Grade. Contrast sensitivity was measured with the CSV-1000, and the Dry Eye Related Quality of Life Score (DEQS) questionnaire was performed at the end of the 2 week study period."

Reviewer 2 Report

The objective of this study is somewhat novel and interesting, especially regarding that it links the relatively new eyedrops (diquafosol) and surgical modality (femtosecond laser-assisted cataracts). This study analyzes the protective effects of diquafosol on the ocular surface two weeks after femtosecond laser-assisted cataract surgery. While the results seem intriguing, the following comments need addressing:

1.     Line 41- The label of the reference should be in front of the period, not behind the end.

Patients and Methods

1.     Line 103-105- Patients with dry eye need to be excluded preoperatively. Patients with dry eye may differ in their effect on the use of diquafosol compared to regular patients and this manuscript is about the prevention of surgery-induced dry eye by diquafosol after cataract surgery. You mentioned in your results, "None of the patients had moderate to severe dry eye prior to surgery, and 3 patients occasionally used artificial tears for symptoms of dry eye." Suggest adding dry eye patients to the exclusion criteria.

2.     Line 111-112- " Both the choice of eye to be treated with diquafosol and the eye to first undergo surgery was randomly chosen." Here it is written that one eye is randomly selected for the use of diquafosol, but in the abstract, it is written that the right eye is used for diquafosol.

3.     Line 138-148- The BUT is an important parameter and the method of measurement should be stated in the method. Whether it is measured using the ocular surface analyser or fluorescein staining and whether the measurement is repeated.

4.     Line 138-148- When conducting statistical analysis, whether the normality and chi-squared test have been performed. Normality and chi-squaredness are normally required to be satisfied using a t-test. Normality needs to be satisfied using the mean (standard deviation).

Results

1. The follow-up period is short, only observed up to 2 weeks after the operation. Whether there will still be an effect on the ocular surface after discontinuation of the drug. A longer follow-up period is recommended..

References

1.The cited references rate within 5 years is low. The authors should increase cited references within 5 years.

Author Response

We thank the Reviewers for the constructive criticism to improve the quality of our paper. The following is a point-for-point response to each comment by the Reviewers.

Patients and Methods

  1. Line 103-105- Patients with dry eye need to be excluded preoperatively. Patients with dry eye may differ in their effect on the use of diquafosol compared to regular patients and this manuscript is about the prevention of surgery-induced dry eye by diquafosol after cataract surgery. You mentioned in your results, "None of the patients had moderate to severe dry eye prior to surgery, and 3 patients occasionally used artificial tears for symptoms of dry eye." Suggest adding dry eye patients to the exclusion criteria.

Response: Thank you for raising an important point. We agree that patients with definitive dry eye should be excluded from the study. We have excluded these three patients, and revised the manuscript as "3 patients who occasionally used artificial tears for symptoms of dry eye were excluded from the study", and also changed the numbers of participants.

  1. Line 111-112- " Both the choice of eye to be treated with diquafosol and the eye to first undergo surgery was randomly chosen." Here it is written that one eye is randomly selected for the use of diquafosol, but in the abstract, it is written that the right eye is used for diquafosol.

Response: We apologize for the confusion. Our trial randomly chose which eye is to use diquafosol and which eye is to undergo surgery using a table of random numbers. The abstract was rewritten as follows; “Patients received 3% diquafosol eyedrops 6 times daily in one eye (diquafosol group), and physiological saline 6 times a day in another eye (control group).” All participants were randomly assigned to two groups using a table of random numbers.

  1. Line 138-148- The BUT is an important parameter and the method of measurement should be stated in the method. Whether it is measured using the ocular surface analyser or fluorescein staining and whether the measurement is repeated.

Response: We revised the text as follows; "BUT was measured by slit lamp with a blue filter using a standard fluorescein paper strip and balanced saline solution." We also modified the last sentence of the paragraph as follows; "Both the BUT and SPK scores were recorded by 2 independent examiners (RK and YS) who were masked to which group the eye belonged."

  1. Line 138-148- When conducting statistical analysis, whether the normality and chi-squared test have been performed. Normality and chi-squaredness are normally required to be satisfied using a t-test. Normality needs to be satisfied using the mean (standard deviation).

Response: Normality and chi-square tests were also performed in this study. For the normality test, the mean (standard deviation) was used. We have added a description of this content to the Method section.

Results

  1. The follow-up period is short, only observed up to 2 weeks after the operation. Whether there will still be an effect on the ocular surface after discontinuation of the drug. A longer follow-up period is recommended.

Response: We totally agree that a longer follow-up period will offer more information. However, the objective of this study was to observe short-term dry eye symptoms due to surgery which may affect patient satisfaction. We added the following sentence in the discussion to explain this point; "The short follow up period of 2 weeks may be a limitation of this study. However, the objective of this study was to observe the effects of diquafosol on short-term dry eye symptoms due to surgery, which may affect patient satisfaction."

References

1.The cited references rate within 5 years is low. The authors should increase cited references within 5 years.

Response: We added the following reference (ref. no. 33) published this year that suggests that dry eye therapy may improve refractive accuracy in cataract surgery.

Teshigawara, T.; Meguro, A.; Mizuki, N. The Effect of Rebamipide on Refractive Accuracy of Cataract Surgery in Patients with Dry Eye. Ophthalmol Ther 2022, 11, 603-611, doi:10.1007/s40123-022-00457-3.

Reviewer 3 Report

I find it a very interesting publication. However, to be suitable for publishing a few minor changes must be made:

- Figure 1 and 2 are very small and the data is not appreciated

- In the discussion it is necessary to explain what was found in the results with other studies

- Also in the discussion add the limitations of the study and future perspectives

Author Response

We thank the Reviewers for the constructive criticism to improve the quality of our paper. The following is a point-for-point response to each comment by the Reviewers.

- Figure 1 and 2 are very small and the data is not appreciated

Response: We enlarged the figures as suggested.

- In the discussion it is necessary to explain what was found in the results with other studies

Response: We added the following reference (ref. no. 33) published this year that suggests that dry eye therapy may improve refractive accuracy in cataract surgery.

Teshigawara, T.; Meguro, A.; Mizuki, N. The Effect of Rebamipide on Refractive Accuracy of Cataract Surgery in Patients with Dry Eye. Ophthalmol Ther 2022, 11, 603-611, doi:10.1007/s40123-022-00457-3.

- Also in the discussion add the limitations of the study and future perspectives

Response: Thank you for the suggestion. We added the following sentence in the Discussion; "The short follow up period of 2 weeks may be a limitation of this study. However, the objective of this study was to observe the effects of diquafosol on short-term dry eye symptoms due to surgery, which may affect patient satisfaction."

Reviewer 4 Report

Dear Authors

Thanks for your hard work.

Some queries from me.

1)    There are studies reporting incidence of dry eye after phacoemulsification that the IOLs were not limited to multifocal, why did you not recruit FLACS with other types of IOL in your study?

2)    “Both the choice of eye to be treated 111 with diquafosol and the eye to first undergo surgery was randomly chosen”, please elaborate which method you use.

3)    Were patients who had been on artificial tear prior to surgery eligible or not to the study?

4)    Please clarify more details on sample size calculation, why it was 23 eyes each group.

5)    Despite the short observation of our study due to COVID pandemic was understandably noted, patients with cataract surgery usually need follow up beyond 2 weeks. I believe that the management during the pandemic would be specific according to region or country. Nonetheless, it would be some exceptions for seeing patients in person when needed such as post -op patients, that it is for patients’ own safety, or in patients who were ongoing in the study. I fully agree that 2 weeks follow up was too short for its clinical outcome to convince the benefit of Diquafosol compared to control. If possible, please clarify more on the reason of your study termination, or please state what follow up period you actually scheduled in your submitted protocol.

Best regards.

Author Response

Response to Reviewers

We thank the Reviewers for the constructive criticism to improve the quality of our paper. The following is a point-for-point response to each comment by the Reviewers.

1)    There are studies reporting incidence of dry eye after phacoemulsification that the IOLs were not limited to multifocal, why did you not recruit FLACS with other types of IOL in your study?

Response: Our study was limited to multifocal IOLs because we wished to compare the effect of Diquafosol on near and intermediate visual acuity following multifocal IOL implantation, since we considered that visual performance of multifocal IOL seems to be more influenced by dry eye. We could not compare data with mono-focal IOLs since we only perform FLACS for multifocal IOLs.

2)    “Both the choice of eye to be treated with diquafosol and the eye to first undergo surgery was randomly chosen”, please elaborate which method you use.

Response: All participants were randomly assigned to two groups using a table of random numbers.

3)    Were patients who had been on artificial tear prior to surgery eligible or not to the study?

Response: Other Reviewers have raised this point, and we have decided to exclude 3 patients who have been on artificial tears prior to surgery. The text was modified as follows in the first paragraph of Results; "This study included 42 eyes of 21 patients (11 male, 10 female). One patient suffered mild age-related macular degeneration (AMD) and 3 patients who occasionally used artificial tears for symptoms of dry eye were excluded from the study."

4)    Please clarify more details on sample size calculation, why it was 23 eyes each group.

Response: Our study design was to use both eyes of the same patient, one eye with diquafosol and one eye with control. Therefore, the number of eyes in each group will always be the same. I hope this answers your question.

5)    Despite the short observation of our study due to COVID pandemic was understandably noted, patients with cataract surgery usually need follow up beyond 2 weeks. I believe that the management during the pandemic would be specific according to region or country. Nonetheless, it would be some exceptions for seeing patients in person when needed such as post -op patients, that it is for patients’ own safety, or in patients who were ongoing in the study. I fully agree that 2 weeks follow up was too short for its clinical outcome to convince the benefit of Diquafosol compared to control. If possible, please clarify more on the reason of your study termination, or please state what follow up period you actually scheduled in your submitted protocol.

Response: Thank you for raising an important point. The objective of this study was to observe short-term dry eye symptoms due to surgery which may affect patient satisfaction. The original protocol of the study was to follow-up for 3 weeks post-operatively, but was shortened to 2 weeks because of the pandemic. We determined that 2 weeks was enough to analyze the effect of diquafosol for immediate post-operative dry eye since other articles suggested the effects of diquafosol were observed within  2 weeks, and that postoperative dry eye is worse a few days after surgery.

Round 2

Reviewer 4 Report

Dear authors

Thanks for your response to the previous review.

Referring to Q 4

4) Please clarify more details on sample size calculation, why it was 23 eyes each group.

Response: Our study design was to use both eyes of the same patient, one eye with diquafosol and one eye with control. Therefore, the number of eyes in each group will always be the same. I hope this answers your question.

May I clarify your understanding on this? In the epidemiology, particularly in the prospective randomized comparison study, the appropriate sample size has to be calculated through a proper method to assure that the number of subjects in each arm is adequate to prove the statistical significance if there truly was (or in true statistic sense, it means to accept or deny the null hypothesis). Your answer just showed that this is the number you anecdotally set.

By the way, you have changed the number of subjects but the results did not change at all, even the decimal point, which is impossible as your source of calculation has not been the same. Please immensely reconsider this, since it did make your study not reliable.

Best regards

Author Response

>In the epidemiology, particularly in the prospective randomized comparison study, the appropriate sample size has to be calculated through a proper method to assure that the number of subjects in each arm is adequate to prove the statistical significance if there truly was (or in true statistic sense, it means to accept or deny the null hypothesis). Your answer just showed that this is the number you anecdotally set.

Response: The number of cases was calculated with error (difference in the effect = 20%, confidence level = 95%, and standard deviation = 20), and the number of cases with a significant difference in the effect between the two groups was 16.

>By the way, you have changed the number of subjects but the results did not change at all, even the decimal point, which is impossible as your source of calculation has not been the same. Please immensely reconsider this, since it did make your study not reliable.

Response: We apologize that we did not recalculate statistics after changing the number of cases to 21. Cases who were treated for dry eye before surgery were omitted as suggested by the reviewers.